# Citizen attitudes towards the environment and association with perceived threats to the countryside: Evidence from countries in five European biogeographic zones

Barbara J. Stewart-Knox[1], Brendan P. Bunting[2], Shan Jin[3]*, Sophie Tindale[4], Victoria Vicario-Modroño[5], Simona Miškolci[6], Mercy Ojo[4], Pedro Sánchez-Zamora[5], Rosa Gallardo-Cobos[5], Paul Newell-Price[7], Martijn Sonnovelt[8], Erik Hunter[9], Lynn J. Frewer[4]*

1 Department of Psychology, School of Social Sciences, University of Bradford, Bradford, United Kingdom, 2 Psychology Research Institute, Ulster University, Coleraine, Northern Ireland, United Kingdom, 3 Faculty of Business and Law, University of Portsmouth, Portsmouth, United Kingdom, 4 School of Natural and Environmental Sciences, Newcastle University, Newcastle upon Tyne, United Kingdom, 5 Department of Agricultural Economics, ETSIAM, Universidad de Córdoba, Córdoba, Spain, 6 Department of Regional and Business Economics, FRDIS, Mendel University in Brno, Brno, Czech Republic, 7 RSK ADAS Ltd, Spring Lodge, Helsby, United Kingdom, 8 World Food System Center, ETH Zurich, Zürich, Switzerland, 9 Department of Work Science, Business Economics and Environmental Psychology, Swedish University of Agricultural Sciences, Uppsala, Sweden

☯ These authors contributed equally to this work.
* andy.jin@port.ac.uk (SJ); lynn.frewer@newcastle.ac.uk (LJF)

## Abstract

Citizens play a crucial role in attaining the United Nations 2030 sustainable development goals (SDGs). There is growing awareness of the importance of understanding citizen perspectives on environmental issues, in relation to developing and maintaining sustainable lifestyles, and in addressing perceived threats to protection and restoration of ecosystems and biodiversity. This analysis sought to understand people's attitudes towards environmental conservation, how they relate to perceived threats to the countryside, and to determine how attitudes and perceived threats vary demographically and between countries. A survey was administered to citizens (quota sampled on age, gender, education, and split between rural and urban residency) across five countries representative of differing biogeographical regions (N = 3,190): Czech Republic (n = 649) (Continental); Spain (Mediterranean) (n = 623); Sweden (Boreal) (n = 645); Switzerland (Alpine) (n = 641); United Kingdom (UK) (Atlantic) (n = 632). Attitudes were measured using the Environmental Attitudes Inventory (EAI-24) on 2 factors (utilization; preservation) and perceived threat to the countryside on 1-factor (15 items). Multigroup regression analysis indicated that preservationist attitudes were associated with greater perceived threat to the countryside in all five countries. Higher perceived threat was associated with activities linked to environmental degradation, socio-economic uncertainty and risks in agri-food supply chains in all countries. The "bad behaviour of visitors" was the greatest perceived threat in the Czech Republic, Switzerland and the UK, while "lack of young farmers taking over farming" was the greatest perceived threat in Spain and Sweden. To promote pro-environmental attitudes and obtain greater public

**Data Availability Statement:** The survey instrument and data are available at https://zenodo.org/records/12819487.

**Funding:** The SUPER-G project (Grant Agreement No.: 774124) has received funding from the European Union Horizon 2020 Research and Innovation Programme. The views and opinions expressed in this paper do not represent the official position of the European Commission and is entirely the responsibility of the authors. The funders had no role in study design, data collection and analysis, decision to publish, or preparation of the manuscript.

**Competing interests:** The authors declare that they have no known competing financial interests or personal relationships that could have appeared to influence the work reported in this paper.

support for policies and interventions targeting environmental conservation, communication about environmental threats is needed, together with threat mitigation measures. Raising peoples' awareness of threats to the countryside through targeted communications could promote pro-environment attitudes and potentially result in pro-environmental behaviours.

## 1. Introduction

There is growing concern about achieving the 2030 targets set out by the United Nations (UN) in achieving the sustainable development goals (SDGs) [1–3]. A scoping review of literature published from 2017–2021 indicated a relative lack of interest among the scientific community in food systems related to SDG 2 zero hunger (food security; sustainable agriculture; improved nutrition) [4]. This is surprising given that SDG 2 is pivotal to attaining SDG 3 (good health and wellbeing), SDG 12 (responsible production and consumption) as well as SDG 15 (protection and restoration of ecosystems and biodiversity) [5], and that trade-offs and synergies between these SDGs are influenced by human perceptions, attitudes and behaviour [6]. These SDGs are all related to human activity, which underlines the importance of understanding citizen perspectives on environmental issues [7].

Instigating and implementing measures to protect the environment, will promote human health and wellbeing, and ensure the sustainability of food production as well as ecological survival [8, 9]. The Millennium Ecosystem Assessment is a UN-sponsored activity aimed at analysing the impact of human actions on ecosystems that describes ecosystem services (ES) as the tangible and intangible benefits humans obtain from both natural ecosystems, and those modified by humans [10]. Rural areas can provide multiple essential ES, including provisioning services (e.g. human food, animal feed and timber production), regulating and maintenance services (e.g. carbon sequestration, erosion control, nutrient cycling, and pollination), and cultural services (e.g. aesthetic value, and tourism) [11, 12]. In recent years, rural areas have seen a move away from agricultural productivism and toward a multifunctional model of land-use that includes leisure and tourism, whilst preserving the cultural ES functions of landscapes and the countryside [13, 14]. While the leisure industry must consider the issue of sustainability in land use [15], people may be unwilling to embrace low-carbon tourism or pay a premium for land access, which could compromise the delivery of regulating and maintenance ES [16]. Achieving sustainable land use will require policies linked to social innovations that will bring about changes in food production, food consumption and in people's lifestyle practices [9]. Understanding citizen attitudes to the environment and land use is therefore important in the design and implementation of effective environmental planning, especially if more sustainable food systems are to be developed, and rural management is to achieve multifunctional rural landscapes.

Historically, human interaction with the environment has been viewed in terms of intrinsic and instrumental value, and only recently has the importance of the 'relational' value of the environment, which encompasses emotional and experiential factors, received attention [17–19]. Schultz defined environmental values in terms of affective (emotional) environmental concern and classified them into three distinct elements; *egoistic* (self-concerned), *altruistic* (others-concerned) and *biospheric* (nature-concerned) [20]. Environmental attitudes also have a strong affective component [21]. Attitudes to the environment can be culturally determined [22] and are integral to an individual's social identity [23]. Accordingly, pro-environmental attitudes are more frequently observed in people who are more altruistic [24], less materialistic

in their world view [25], and those with more liberal political views [26]. Environmental attitudes therefore are complex and multifaceted and, as such, subject to individual differences that may need to be addressed in policies and intervention strategies to promote pro-environmental attitudes and, hence, behaviours.

Existing research indicates wide variation in public attitudes towards the countryside [27] including between regions and demographic groups [28]. Attitudes tend to be more pro-environment among women (Note a recent publication reported no gender difference [92].) [29–31], those who have spent longer in education [29, 30], and those of higher socio-economic status (SES) [31]. Evidence for age differences in environmental attitudes is less consistent with some studies reporting more positive attitudes towards environmental conservation among older people [32, 33], and others in younger people [34–36]. Existing evidence implies that urban and rural dwellers hold very different views with urban dwellers holding more positive environmental attitudes [34, 36]. *A priori* qualitative research conducted as part of the Super-G project [28] indicated that even when people in different countries agreed that farming for biodiversity was preferable to managing conventional agricultural practices within landscapes, rural and urban dwellers held contrasting perspectives on the prioritisation of ES. Rural dwelling participants perceived that more complex groupings of ESs could be obtained from the countryside compared to urban dwellers. It is therefore important for policies and interventions to consider environmental attitudes between different social groups, rural *versus* urban residency, and across different countries with different regulatory and biogeographic contexts.

Although more positive 'general' environmental attitudes have been linked to specific pro-environmental behavioural intentions [37] and behaviours [22, 26, 28, 38–40], this relationship is not always observed [26, 41]. Protection Motivation Theory (PMT) seeks to explain how people react in the face of a perceived threat and assumes that behaviour is a function of motivation to behave in a way that mitigates a perceived threat (taking account of and individual's threat appraisal and coping appraisal) [42, 43]. It is possible therefore that pro-environment attitudes may not always translate into actual pro-environmental behaviour because the perceived environmental threat is context-dependent and a function of the expectation that a behaviour will lead to a specific outcome. PMT considers the perceived value or utility of such an outcome [42]. For example, people may be more favourable towards electric vehicles where the costs of a conventional vehicle (a perceived negative consequences) are perceived to be greater [43]. Similarly, people's motivation to engage in re-cycling behaviours may be associated with their perceived higher cost of not re-cycling [44]. In the context of biodiversity, an individual's propensity for action to control invasive species has been linked to their experiences of biodiversity losses [45]. Environmental attitudes therefore may vary according to the characteristics and associated perceived costs of different environmental threats. Environmental threats that are perceived as greater in magnitude have been associated with more positive general environmental attitudes [46], a greater propensity to engage in behaviours which mitigate negative environmental impacts [47], and greater support for legislation which is directed towards mitigating climate change impacts [48]. Perception of environmental threats can also change over time and in the light of new information [49]. Thus, increasing people's awareness about potential threats to the environment might result in a shift towards more pro-environmental attitudes and hence greater support for pro-environmental policy measures [50, 51]. Together this suggests that perceived threat is relevant to understanding environmental attitudes and in framing communication and policies to manage environmental behaviour.

This research meets a need for data-driven models of citizen perspectives on environmental sustainability [52, 53]. The objective has been to understand environmental attitudes and their association with ratings of perceived environmental threats related to the agri-food sector

(SDG 2 * food security; sustainable agriculture), awareness of which could be heightened through public communication channels and other policy measures. The aim of this novel analysis will be to inform the development of effective policies and interventions to encourage pro-environmental attitudes and behaviours, through consideration of perceived environmental threats that could potentially influence whether attitudes towards the environment translate into behaviour. The results will provide a starting point for planning and managing interventions to motivate people to protect the environment. Based on the results, it should be possible to tailor communications about environmental threats to align with preservationist and utilitarian attitudes, and to target interventions (for example, widening access to rural areas) to socio-demographic groups within each of the five countries included in this research.

To the authors' knowledge, previous research has not considered the relationship between environmental attitudes and perceived environmental threats to the countryside, nor has there been any comparative analysis of environmental attitudes in large representative samples and in different socio-economic groups. This research has considered environmental attitudes and perceived threats cross-nationally in countries representing different geographical areas (the Czech Republic, Spain, Sweden, Switzerland and United Kingdom). Environmental attitudes have been assessed using the validated Environmental Attitudes Inventory (EAI-24), which considers 'preservationist' attitudes (the extent to which people perceive that preserving and maintaining ecosystems and the diversity of natural species is important), and 'utilitarian' attitudes (the belief that using and altering ecosystems and natural species for human objectives is right, appropriate and necessary) [54–56]. Items which have assessed perceived environmental threats have validity given they have been drawn from *a priori* qualitative research [28]. The analysis has sought to determine how utilitarian and preservationist environmental attitudes relate to perceptions of specific environmental threats. It is predicted that pro-environmental attitudes will be positively associated with greater levels of perceived environmental threats, and that this association will vary by age and between genders, education level, and between urban and rural dwellers.

## 2. Methods

This secondary analysis draws on survey data collected as part of the European Horizon 2020 funded SUPER-*G* project. The research compares environmental attitudes and perceived threat to the countryside cross-sectionally and the relationships therein across five European countries and in different demographics. Ethical approval for the survey research was granted by Newcastle University, Faculty of Science, Agriculture and Engineering Ethics Committee on 21/08/2020 [Ref 20-TIN-029]. The survey instrument and data are available at https://zenodo.org/records/12819487.

### 2.1 Sampling and procedure

Sample recruitment and survey data collection were undertaken by a social research agency (Qualtrics LLC) across five countries (N = 3190): the Czech Republic (n = 649); Spain (n = 623); Sweden (n = 645); Switzerland (n = 641); United Kingdom (UK) (n = 632). These countries were selected to represent the Continental, Mediterranean, Boreal, Alpine and Atlantic biogeographical regions respectively. In Europe, the Alpine region (Switzerland) and the Boreal region (Sweden) are considered to have the best conservation status for habitats and native species, whereas the Atlantic region (the UK) and the Continental region (Czech Republic) have the worst conservation status [57]. The Mediterranean region (Spain) has the largest area of degraded forests, grasslands, scrub and heath. Variations in environmental conditions across regions may lead to differences in citizens' environmental attitudes and perceived environmental threats to the countryside [58].

Participants were quota sampled to be representative of national populations on age, gender (50%), socio-economic status, education level and 50% rural *versus* urban residency. Data were collected using online survey conducted between the 1st October and the 1st November, 2021. Following initial recruitment, participants were provided with an information sheet outlining the aims of project, how collected data would be managed and used, and which informed them of their right to discontinue or withdraw from the survey at any time. Subsequently, participants were presented with a consent question. Only those who selected 'yes', indicating they had read the information sheet and agreed to take part in the survey, were able to proceed.

## 2.2 Materials

**2.2.1 The environmental attitudes inventory.**   The environmental attitudes inventory (EAI) is a psychometric tool that measures attitudes representative of instrumental, intrinsic and relational aspects of the environment and that considers both productivist and conservational attitudes. It also considers environmental attitudes that tap into social identity. The original environmental attitudes inventory (EAI) comprised twelve scales [56]. The shorter 24-item environmental attitudes inventory (EAI-24) has been used in this research. The EAI-24 has been previously administered in Spain [59], Portugal [32] and France [54], and has been found to have similar psychometric properties to the original version. It has demonstrated satisfactory predictive validity and test-retest reliability [54]. The EAI appears to have good discriminate validity in that older people, women and natural scientists have been found to score higher on preservation and lower on utilisation [32]. All items were prompted by *'How far do you agree or disagree with the following statements'* and scored on a five-point Likert scale ranging from *strongly agree* to *strongly disagree*. Negative items were reverse scored.

**2.2.2 Perceived threat to the countryside.**   Questionnaire items relating to perceived threat [15] were informed by the findings of the *a priori* qualitative research investigating public attitudes and perceptions of grassland landscapes [28]. The wordings were derived from focus group discussion dialogue tapping into perceived threat to the countryside. A systematic literature review of 51 studies concluded that threats to the countryside tended to be concerned with land use, land management, social attitudes, industrial development, human recreation and tourism [8], issues which were subsequently included as items in the questionnaire. All items were prompted by *'How far do you agree or disagree with the following statements on the causes of problems in the countryside'* for which responses were on a five-point Likert-scale ranging from *strongly agree* to *strongly disagree*.

**2.2.3 Exogenous factors.**   Demographic factors assessed were age; gender (male/female/ prefer not to say); residence (urban/rural); education (4 levels: secondary education or less; upper secondary education; undergraduate degree or diploma; postgraduate degree or qualification). The frequency with which respondents visited the countryside was also recorded.

## 2.3 Data analysis

Environmental attitudes were entered into the analysis as two factors with good reliability '*preservationist*' ($\alpha$ = .70) and '*utilitarian*' ($\alpha$ = .82). The outcome variable was the single factor related to perceived threat to the countryside, which also had good reliability ($\alpha$ = .84). The general factor for perceived threats to the environment was then regressed onto both environmental attitudinal factors (utilitarian and preservationist) and the five exogenous measures: age (continuous); gender (male/female); educational—4 levels (secondary education or less; upper secondary education; undergraduate degree or diploma; postgraduate degree or

qualification); location (urban/rural); and frequency of visits to the countryside (continuous). Data were analysed using Mplus Version 8.9 [60].

**2.3.1 Exploratory factor models.** An exploratory factor analysis (EFA) using an oblique geomin factor rotation was conducted on the EAI-24 items using Mplus, initially with UK data only [61–64]. A 5-factor solution was obtained (Table 1). Given the ordinal nature of the response scale (Likert), the estimation method used was robust weighted least squares. The first 2 factors appeared to represent 'utilitarian' and 'preservationist' attitudes to the environment. The remaining 3 factors, which could be seen as relating to activism, scientism and pro-creation, each contained only two items (factor loading greater than 0.4). Four of these 6 items cross-loaded onto either the 'utilitarian' or 'preservationist' factor. To simplify the model, and because of the sparseness of items loading exclusively onto the latter three factors, only items relating to the first two factors were used in the subsequent analysis. These items were selected on the basis that (a) they had factor loadings above 0.4 and (b) they were clearly related to only one factor. This resulted in eleven items on the first factor 'utilitarian' and seven items on the second factor 'preservation' (Table 1). The 'utilitarian' and 'preservationist' factors were

**Table 1. Environmental attitudes inventory (EAI-24) factor structure—UK data (N = 632).**

| | F1 Utilisation | F2 Preservation | F3 Activism | F4 Science | F5 Procreation |
|---|---|---|---|---|---|
| 1. I really like going on trips into the countryside e.g., to forests or fields | -0.116* | 0.389* | -0.141* | -0.006 | -0.290* |
| 2. Protecting peoples' jobs is more important than protecting the environment | 0.596* | -0.043 | 0.077* | -0.172* | -0.050 |
| 3. I'd much prefer a garden that is well groomed and ordered to a wild and natural one | 0.666* | 0.195* | 0.047 | -0.232* | -0.077 |
| 4. I would not get involved in an environmental organisation | 0.482* | -0.054 | 0.017 | -0.642* | 0.087* |
| 5. We need to keep rivers and lakes clean to protect the environment and NOT as places for people to enjoy water sports | -0.002 | 0.573* | -0.112* | -0.112* | -0.125* |
| 6. A married couple should have as many children as they wish as long as they can adequately provide for them | 0.436* | 0.001 | 0.068* | 0.029 | -0.817* |
| 7. I do not believe the environment has been severely abused by humans | 0.611* | -0.151* | 0.019 | -0.102* | 0.072* |
| 8. It makes me feel sad to see forests cleared for agriculture | -0.129* | 0.625* | -0.004 | 0.010 | -0.074 |
| 9. Modern science will NOT be able to solve our environmental problems | 0.033* | 0.615* | 0.936* | -0.010 | 0.047* |
| 10. Grass and weeds growing between pavement stones really looks untidy | 0.630* | 0.231* | -0.061 | -0.380* | -0.152* |
| 11. I do NOT believe humans were created or evolved to dominate the rest of nature | -0.217* | 0.549* | 0.063 | -0.136* | 0.224* |
| 12. Governments should control the rate at which raw materials are used to ensure that they last as long as possible | -0.048 | 0.683* | -0.032 | 0.033 | -0.054 |
| 13. One of the most important reasons to keep lakes and rivers clean is so people have a place to enjoy water sports | 0.684* | 0.114* | -0.093* | 0.162* | 0.046 |
| 14. Modern science will solve all our environmental problems | 0.758* | 0.028* | -0.736* | -0.009 | 0.080* |
| 15. I think spending time in nature is boring | 0.639* | -0.044 | 0.062 | 0.154* | 0.396* |
| 16. I would like to join and actively participate in an environmentalist group | 0.063 | 0.350* | 0.024 | 0.713* | 0.017 |
| 17. Whenever possible I try to save natural resources | -0.020 | 0.577* | -0.099* | 0.222* | -0.163* |
| 18. It does NOT make me sad to see natural environments destroyed | 0.547* | -0.089* | 0.093 | 0.079 | 0.370* |
| 19. Human beings were created or evolved to dominate the rest of nature | 0.757* | -0.168* | -0.032 | 0.195* | -0.097* |
| 20. Protecting the environment is more important than protecting peoples' jobs | -0.012 | 0.580* | -0.041 | 0.226* | 0.234* |
| 21. Humans are severely abusing the environment | -0.271* | 0.670* | 0.115* | 0.046 | -0.024 |
| 22. I am opposed to governments controlling and regulating the way raw materials are used to make them last longer | 0.500* | -0.034 | 0.065 | 0.039 | 0.003 |
| 23. Families should be encouraged to limit themselves to two children or less | 0.028 | 0.462* | -0.035 | 0.010 | 0.795* |
| 24. I am NOT the kind of person makes effort to conserve natural resources | 0.526* | -0.196* | 0.093* | -0.099* | 0.218* |

*Significant at $P < .05$ level

negatively correlated within all countries (Czech Republic: Est = -0.682, SE = .046, $P < .001$; Spain: Est = -0.642, $SE$ = .046, $P < .001$; Sweden: Est = -0.683, SE = .048, $P < .001$; the UK: Est = -0.632, SE = .037, $P < .001$) except in Switzerland (Est = -0.101, $SE$ = .062, $P$ = 0.104).

A similar approach was adopted for the items assessing perceived threats to the countryside. Initial EFA using an oblique geomin rotation suggested 4 factors. One of these factors was omitted from the analysis as it contained only two items. This left three factors. However, when this three-factor model (having excluded the two items which loaded onto the omitted factor) was fitted to the other countries, it became increasingly difficult to obtain a convergent model with measurement invariance across all the countries. This was deemed impractical, given the lack of items representing the various factors, the extent of the cross-factor loadings and the correlated thresholds. A single-factor solution for these 15 items was assumed to be the optimal strategy in the current context given the measures representing sub-factors across the five societies may be variant (Table 2).

The one-factor model had factor loadings between 0.45 and 0.74, apart from item three 'conversion of pasture or meadow to forest or woodland', which had a factor loading of 0.132 (SE = 0.034) in the exploratory evaluation (Table 2). The factor loading for item 3 increased to above 0.4 in the confirmatory framework. In adopting the one factor approach a multiple-indicator, multiple-cause model (MIMIC) modelling strategy could be utilised [65, 66]. This enabled disparities between responses to the various items are occurring across the countries to be determined (details see S1 File). Through using correlated thresholds, the identity of the various country specific factors could also be obtained. To facilitate a comparative assessment of the perceived threats to the countryside in the five countries, a descriptive analysis of perceived threat ratings was conducted. Inferential statistical analyses, including ANOVA, followed by Tamhane's T2 *post hoc* tests, were then performed to examine variation in citizens' perceptions of threats based on the sum of the factor loadings within the five countries.

**2.3.2 Multi-group analysis.** Given the categorical nature of the observed variables on the perceived threats to the countryside factor, and both attitudinal factors (utilitarian and preservationist), polychoric correlations were computed along with a weighted least squares mean

**Table 2. Geomin rotated factor loadings for perceived threats to the countryside factor UK data (N = 632).**

| Item | Factor Loading |
|---|---|
| Q1. Conversion to urban land use | 0.520* |
| Q2. Conversion of pasture or meadow to cropland | 0.449* |
| Q3. Conversion of pasture or meadow to woodland | 0.132* |
| Q4. Abandonment of land by farmers | 0.477* |
| Q5. Too many livestock causing damage to land | 0.558* |
| Q6. Increased number of visitors (tourism) | 0.554* |
| Q7. Bad behaviour of visitors | 0.711* |
| Q8. Poor farming practices | 0.788* |
| Q9. Misuse of chemical fertilisers | 0.796* |
| Q10. Effects of climate change | 0.680* |
| Q11. Changing diets (eating more/less animal products) | 0.623* |
| Q12. Changing demand for food | 0.624* |
| Q13. Farmers unable to make a living from the land | 0.684* |
| Q14. Lack of young farmers taking over farming | 0.723* |
| Q15. Change to market prices of farm products | 0.735* |

*Significant at $P < .05$ level

and variance (WLSMV) model of estimation and a probit regression model. This inferential analysis was conducted initially on the UK data only [61, 67]. Once a model fit was obtained, the UK factor structure was then imposed on all five countries using a multi-group factorial model [68]. Thus, within each of the measurement models (factors), the factor loadings and thresholds were held equal across the five countries. The models were then tested *via* an examination of the modification indices and the fit indices [69] which provided a test of both factorial invariance (factor loadings) and scalar invariance (thresholds). Factor means were also examined for the unconditioned factors relating to (a) perceived threats to the countryside and (b) environmental attitudes [70].

**2.3.3 Testing the structural model within a multi-group strategy.** The key outcome measure was the factor relating to perceived threat to the countryside. This factor was regressed onto the two attitudinal factors (preservationist and utilitarian) relating to the environment. These attitudinal factors were then regressed onto five exogenous measures (age, gender [man/woman], residence [urban/rural], education level [secondary education or less; upper secondary education; undergraduate degree or diploma; postgraduate degree or qualification], and frequency of visits to countryside). The key outcome measure 'perceived threat to the countryside' was modelled within a confirmatory one-factor framework for the UK dataset. As a first step, a series of correlated thresholds were introduced based on information from the modification indices and through examination of the content of the items. This resulted in the introduction of nineteen correlated residuals/thresholds. These were then retained in all later models. Seven direct effects, two representing attitudinal factors (mediating measures) and five exogenous measures were introduced onto the perceived threats to the countryside factor. The five exogenous measures also had a direct effect onto the two attitudinal (mediation) factors. In addition to these direct and indirect effects on the perceived threat to the countryside factor, several items (3,9,4,7,10) were regressed onto the attitudinal factors. This provided a test of item differential functioning of attitudinal factors on specific perceived threat to the countryside items, over and above the outcome measure of perceived threat to the countryside factor [71–73]. An additional six cross-factor loadings were then introduced within the attitudinal factors as indicated by the exploratory analysis. This model provided a good fit for the UK data (RMSEA: Est = 0.047, 90% CI = 0.042 0.052; CFI = 0.952; SRMR = 0.075).

The model created to explain UK data was then compared with data from Czech Republic, keeping the factor loadings equal between the countries (factorial invariance). This model provided a reasonable description of data in both countries. The adjustments made related to one additional correlated threshold in the perceived threat to the countryside factor, which occurred between items relating to 'increased number of visitors' and 'bad behaviour of visitors'. Two observed items relating to abuse of the environment 'I do not believe the environment has been severely abused by humans' and 'humans are severely abusing the environment' on the preservation factor were also found to be correlated in data from the Czech Republic. In the UK model, which was used as the baseline (comparator) model, there were five direct effects from the attitudinal factors to items relating to perceived threats to the environment. In data from the Czech Republic, six further direct effects were required to go onto perceived threat to the countryside items, five (5, 8, 10, 14, 15) from the utilisation factor and one from the preservationist factor, onto item 3 of perceived threat to the countryside (conversion of pasture or meadow to forest or woodland). With these adjustments to the Czech Republic data, criteria for a good model fit were met (RMSEA: Est = 0.041, 90% CI = 0.038 0.045; CFI = 0.951; SRMR = 0.073).

The model now included the baseline information from the UK model with additional adjustment required in Czech Republic. At this point, data from Switzerland were introduced. These data required an additional correlated threshold relating to the perceived threat item

'too many livestock causing damage to land' within the perceived threat to the countryside factor. Two additional cross-factor loadings were introduced, one 'governments should control the rate raw materials are used' (item 12) onto the utilisation factor and the other 'prefer a garden that is well-groomed and ordered to a natural one' (item 3) onto the preservationist factor. The preservationist factor also influenced item 8 (poor farming practices) of the perceived threat to the countryside factor. At this stage, it was concluded that these data provided a good description of these data (RMSEA: Est = 0.041, 90% CI = 0.038 0.044; CFI = 0.950; SRMR = 0.072). The test of measurement invariance now held for three countries, with only some further minor adjustments required to factor cross-loadings and residual covariances.

Data from Sweden were next introduced into the three countries model. With these additional data, one-cross-factor loading was added from the utilisation factor to the item relating to 'feelings of sadness at the clearing of forests (item 8). This utilisation factor had a direct effect on the perceived threat to the countryside item relating to item 12 'changing demand for food'. For the first time, an item with a different response pattern was identified. This differing item was 'keeping rivers and lakes clean and not for sport' (item 5 from the preservation factor). With these changes the model was taken as an appropriate description of these data (RMSEA: Est = 0.04, 90% CI = 0.037 0.042; CFI = 0.948; SRMR = 0.073).

In the Spanish sample, item 5 of the preservation factor 'keeping of rivers and lakes clean and not for sport', showed response patterns that were different from the other countries. An additional threshold change was also required for item 3 'prefer a garden that is well-groomed and ordered to a wild and natural one'. Four of the items (2,13,14,15) relating to perceived threats to the environment had a differential response over and above what could be explained by the direct effect from the attitudinal domains. Of the four items with this differential response, three were affected by the preservation factor and one (item 2) by the utilisation factor. Four additional cross-factor loadings were therefore required along with the attitudinal factors. Three of these cross-factor loadings were from the preservation factor (3,10,13) and the other (item 20) was from the utilisation factor. With these changes, a good fit for these data was achieved (RMSEA: Est = 0.04, 90% CI = 0.038 0.042; CFI = 0.948; SRMR = 0.073). These variables were then entered into the model (Fig 1).

## 3. Results

### 3.1 Sample description

The total sample comprised 3,184, of whom 51% were men and 49% women with a mean age of 45 years (Mean = 45.67; SD = 1.63) 52% of whom lived in urban and 48% in rural areas. Table 3 provides a sample breakdown (including education level) by country.

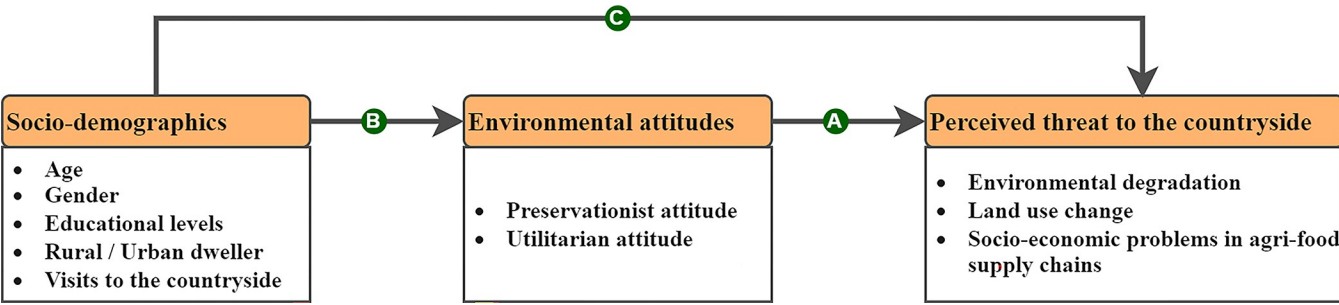

**Fig 1. Environmental attitudes (EAI-24), endogenous variables and perceived threat to the countryside (PTC) plan for analysis.** (A) Arrow A represents the assumed association between environmental attitudes and perceived threat to the countryside; Arrow B represents the assumed association between socio-demographics and environmental attitudes; Arrow C represents the assumed association between socio-demographics and perceived threat to the countryside.

**Table 3. Sample characteristics (percentages) by country, gender, educational level and residency location.**

| Country (N) | | Gender % | | Residency % | | Education level % | | | |
|---|---|---|---|---|---|---|---|---|---|
| | | Man | Woman | Rural | Urban | Secondary or less | Upper secondary | Undergrad degree or diploma | Postgrad degree or qualification |
| UK | 632 | 48 | 52 | 49 | 51 | 20 | 38 | 33 | 9 |
| Czech Republic | 649 | 50 | 50 | 50 | 50 | 9 | 70 | 19 | 2 |
| Switzerland | 638 | 45 | 55 | 45 | 55 | 7 | 51 | 20 | 23 |
| Sweden | 643 | 57 | 43 | 48 | 52 | 21 | 34 | 32 | 13 |
| Spain | 622 | 57 | 43 | 46 | 54 | 35 | 27 | 29 | 9 |

### 3.2 Environmental attitudes inventory

There were significant between country differences in preservationist (F(4) = 8.081; $P < .001$) and utilitarian (F(4) = 17.677; $P < .001$) attitudes, with small effect sizes ($ETA^2 = .010$; $ETA^2 = .022$, respectively) (Table 4). Preservationist attitudes were higher in Spain than in Czech Republic ($P = .002$) or Switzerland ($P < .001$) and higher in the UK than in Czech Republic ($P = .023$) or Switzerland ($P < .001$). Preservationist attitudes were lower in Switzerland than in Spain ($P < .001$), Sweden ($P = .015$) or the UK ($P < .001$). Utilitarian attitudes were higher in Switzerland than in the other countries ($P < .001$). Utilitarian attitudes were lower in Sweden than in Switzerland ($P < .001$), the UK ($P = .007$) or Spain ($P = .032$).

### 3.3 Perceived threat to the countryside

There were significant between country differences in perceived threat to the countryside (F(4) = 12.693; $P < .001$), with a small effect size ($ETA^2 = .016$). Perceived threat to the countryside was higher in the UK than in Czech Republic ($P < .001$), Spain ($P < .001$), Sweden ($P < .001$) or Switzerland ($P < .001$) (Table 4). Among the five countries, higher perceived threat tended to be associated with activities linked to environmental degradation (especially 'bad behaviour of visitors' and 'misuse of chemical fertilizers') and socio-economic uncertainty and risks related to agri-food supply chains (especially 'lack of young farmers taking over farming' and 'change to market prices of farm products') compared to improper land use (such as 'conversion to urban land use' and 'conversion of pasture or meadows to crop land') (Table 5). Specifically, highest perceived threat was associated with 'bad behaviour of visitors' in Czech Republic, Switzerland and the UK, and with 'lack of young farmers taking over farming' in Spain and Sweden.

**Table 4. Perceived threat to the countryside, preservationist and utilitarian environmental attitudes in five countries.**

| Country (N) | | Preservationist | | Utilitarian | | Perceived Threat to the Countryside | |
|---|---|---|---|---|---|---|---|
| | | *Mean* | *SD* | *Mean* | *SD* | *Mean* | *SD* |
| United Kingdom | *632* | 3.8871 [a] | .02540 | 2.3504 [b] | .77386 | 3.6261 [a] | .57163 |
| Czech Republic | *649* | 3.7784 [b c] | .02490 | 2.2767 [b c] | .67233 | 3.4112 [b] | .58398 |
| Switzerland | *641* | 3.7434 [c] | .02602 | 2.5400 [a] | .86510 | 3.4241 [b] | .62722 |
| Sweden | *645* | 3.8597 [a b] | .02565 | 2.2093 [c] | .70917 | 3.4991 [b] | .55977 |
| Spain | *623* | 3.9157 [a] | .02660 | 2.3258 [b] | .69798 | 3.4850 [b] | .67620 |
| Total Sample | *3190* | 3.8362 | .01155 | 2.3402 | .75465 | 3.4885 | .60926 |

[a–c] Values with the same letter as superscript indicate not significantly different means, and different superscripts indicate significantly different means between the segments, following ANOVA *post hoc* tests (Tamhane's T2) at $p < 0.05$.

**Table 5. Perceived threat of specific countryside issues by country (means and standard deviations) (N = 3190).**

| Type of threat | Item | Czech Republic | Spain | Sweden | Switzerland | UK | Total |
|---|---|---|---|---|---|---|---|
| Improper land use by rural communities | Q1 | 3.320(1.265) | 3.136(1.408) | 3.569(1.160) | 3.028(1.401) | 3.701(1.144) | 3.351(1.304) |
| | Q2 | 3.242(1.052) | 3.289(1.049) | 3.278(0.974) | 3.092(1.020) | 3.432(0.931) | 3.266(1.011) |
| | Q3 | 3.034(0.981) | 3.347(1.100) | 3.222(1.001) | 3.081(1.109) | 3.078(1.008) | 3.151(1.047) |
| | Q4 | 3.059(1.218) | 3.013(1.399) | 3.403(1.101) | 3.164(1.136) | 3.323(1.079) | 3.193(1.200) |
| | Q5 | 2.778(1.011) | 2.981(1.249) | 2.947(1.091) | 3.335(1.109) | 3.354(1.048) | 3.078(1.126) |
| Activities linked to environmental degradation | Q6 | 3.320(1.061) | 3.445(1.145) | 3.307(1.045) | 3.314(1.104) | 3.698(0.875) | 3.415(1.060) |
| | Q7 | 3.926(1.083) | 3.778(1.339) | 3.873(1.093) | 3.763(1.008) | 4.049(1.063) | 3.878(1.126) |
| | Q8 | 3.663(1.031) | 3.485(1.328) | 3.433(1.079) | 3.335(1.025) | 3.731(1.024) | 3.529(1.112) |
| | Q9 | 3.824(1.133) | 3.729(1.343) | 3.847(1.099) | 3.741(1.090) | 3.968(1.059) | 3.822(1.151) |
| | Q10 | 3.581(1.042) | 3.790(1.241) | 3.719(1.072) | 3.710(1.019) | 3.834(1.024) | 3.726(1.085) |
| Socio-economic uncertainty and risks in agri-food supply chains | Q11 | 3.169(0.970) | 3.474(1.078) | 3.231(1.042) | 3.454(1.004) | 3.426(0.927) | 3.349(1.012) |
| | Q12 | 3.311(0.907) | 3.560(0.996) | 3.350(0.892) | 3.465(0.938) | 3.592(0.862) | 3.454(0.926) |
| | Q13 | 3.348(1.013) | 3.300(1.234) | 3.760(0.926) | 3.569(0.999) | 3.706(0.963) | 3.537(1.047) |
| | Q14 | 3.867(0.950) | 4.026(1.027) | 3.918(0.913) | 3.696(1.011) | 3.791(0.906) | 3.859(0.968) |
| | Q15 | 3.724(0.855) | 3.923(1.024) | 3.631(0.883) | 3.615(0.970) | 3.709(0.891) | 3.719(0.932) |

## 3.4 Association between environmental attitudes and perceived threat to the countryside

Based on polychoric correlations, direct associations between preservationist and utilization environmental attitudes inventory (EAI) factors and perceived threats to the countryside were identified (Table 6). Mediating effects of preservationist and utilization environmental attitudes on the relationship between socio-demographics and perceived threats to the countryside were also observed (see S1 Table). The key findings are presented in Table 7.

**Table 6. Structural model results (unstandardised) comparing direct associations between preservationist and utilization environmental attitudes inventory factors and perceived threats to the countryside and with exogenous variables by country (N = 3184).**

| | UK | | Czech Republic | | Switzerland | | Sweden | | Spain | |
|---|---|---|---|---|---|---|---|---|---|---|
| | EST | SE | EST | SE | EST | SE | EST | SE | EST | SE |
| Utilization + PTC | 0.097 | 0.070 | 0.239 | 0.057** | 0.274 | 0.051** | 0.287 | 0.098** | 0.442 | 0.108** |
| Utilization + Gender | -0.202 | 0.086* | 0.237 | 0.110* | 0.346 | 0.117** | 0.117 | 0.066 | 0.331 | 0.108** |
| Utilization + Age | -0.012 | 0.003** | -0.006 | 0.003 | -0.009 | 0.003** | -0.011 | 0.002** | -0.014 | 0.004** |
| Utilization + Education | -0.138 | 0.050** | -0.112 | 0.085 | 0.285 | 0.068** | -0.019 | 0.032 | -0.139 | 0.065* |
| Utilization + Urban/Rural | -0.007 | 0.113 | 0.078 | 0.148 | 0.054 | 0.129 | -0.037 | 0.076 | 0.004 | 0.126 |
| Utilization + Visits | -0.051 | 0.033 | 0.013 | 0.050 | -0.042 | 0.054 | -0.036 | 0.025 | -0.021 | 0.033 |
| Preservation + PTC | 0.642 | 0.103** | 0.916 | 0.207** | 0.989 | 0.212** | 0.747 | 0.164** | 1.146 | 0.265** |
| Preservation + Gender | 0.149 | 0.073* | -0.090 | 0.053 | -0.076 | 0.055 | -0.128 | 0.067 | -0.183 | 0.061** |
| Preservation + Age | 0.005 | 0.002* | 0.001 | 0.002 | 0.003 | 0.002 | 0.005 | 0.002* | 0.005 | 0.002* |
| Preservation + Education | 0.091 | 0.043* | -0.043 | 0.042 | 0.000 | 0.027 | 0.002 | 0.030 | 0.020 | 0.035 |
| Preservation + Urban/Rural | 0.069 | 0.093 | -0.001 | 0.070 | 0.120 | 0.064 | 0.054 | 0.072 | 0.038 | 0.072 |
| Preservation + Visits | 0.062 | 0.028* | -0.008 | 0.025 | 0.056 | 0.026* | 0.038 | 0.026 | 0.017 | 0.019 |
| PTC + Gender | 0.057 | 0.062 | 0.003 | 0.060 | -0.034 | 0.064 | -0.199 | 0.065** | -0.045 | 0.109 |
| PTC + Age | 0.003 | 0.002 | -0.003 | 0.002 | 0.001 | 0.002 | 0.004 | 0.002* | 0.009 | 0.005* |
| PTC + Education | 0.033 | 0.032 | 0.145 | 0.052** | 0.030 | 0.038 | 0.026 | 0.030 | 0.088 | 0.064 |

**Table 7. Key findings of the multi-group analysis.**

| Themes | | Results | Summary |
|---|---|---|---|
| **Environmental attitudes** | Socio-demographic predictors of environmental attitudes | *Age*: Age was a stable positive predictor of people's preservationist attitude in the UK, Spain and Sweden and a negative predictor of utilitarian attitudes in all countries except the Czech Republic).<br>*Gender*: In the UK, women had a stronger tendency towards preservationist attitude and a weaker tendency towards utilitarian attitude compared to men. Women showed a stronger tendency towards utilitarian attitude in the Czech Republic, Switzerland and Spain, and a weaker tendency towards preservationist attitudes in Spain.<br>*Education*: More educated people in the UK had a stronger tendency towards preservationist attitudes and a weaker tendency towards utilitarian attitudes compared with the less educated. The more educated in Spain also had a weaker tendency towards utilitarian attitudes compared with the less educated. In contrast, the more educated in Switzerland had a stronger tendency towards utilitarian attitudes.<br>*Frequency of visiting the countryside*: Those who visit the countryside more frequently had a stronger tendency towards preservationist attitudes in Switzerland and the UK. | Older people tended towards a preservationist attitude (except in the Czech Republic and Switzerland), while younger people tended towards a utilitarian attitude (except in the Czech Republic).<br>Associations between gender/education and environmental attitudes varied within different countries.<br>Living in rural or urban areas had no association with people's environmental attitudes.<br>Environmental attitudes were less likely to be associated with socio-demographics in the Czech Republic and Sweden than in other countries. |
| **Perceived threats to the countryside** | Socio-economic predictors of perceived threats to the countryside | *Age*: Older people tended to perceive a higher level of threat to the countryside than younger people in Spain and Sweden.<br>*Gender*: In Sweden, women tended to perceive lower level of threat to the countryside compared to men.<br>*Education*: More educated people perceived a higher level of threat to the countryside than the less educated in the Czech Republic. | There were differences in perceived threat to the countryside across socio-demographic groups within countries. |
| | Environmental attitudes as predictors of perceived threats to the countryside | People with a stronger tendency towards either preservationist attitude or utilitarian attitude tended to perceive a higher level of threat to the countryside in all countries (except the UK). Perceived threat to the countryside was more strongly associated with preservationist than utilitarian attitudes. | Preservationist attitude was a stable positive predictor of people's perceived threat to the countryside across five countries. |
| **Mediating effect of environmental attitudes on the relationship between socio-demographics and Perceived threats to the countryside** | Preservationist attitude as a mediator | *Age*: Age (being older) had an indirect positive association with people's perceived threat to the countryside through preservationist attitudes in Spain, Sweden and the UK.<br>*Gender*: Being a woman had indirect positive and negative associations with perceived threat to the countryside through preservationist attitude in the UK and Spain, respectively.<br>*Education*: Being more educated had an indirect positive association with people's perceived threat to the countryside through preservationist attitude in the UK.<br>*Frequency of visiting the countryside*: More frequent visits to the countryside had an indirect positive association with people's perceived threat to the countryside through preservationist attitude in the UK and Switzerland. | Age had both direct and indirect associations with perceived threat to the countryside in Spain and Sweden, and only had an indirect association through environmental attitudes in Switzerland and the UK.<br>Gender had an indirect association with perceived threat to the countryside through environmental attitudes in Czech Republic, Spain, Switzerland and the UK, and had a direct correlation in Sweden.<br>Age and gender had indirect associations with perceived threat to the countryside through environmental attitudes in different countries. |
| | Utilitarian attitude as a mediator | *Age*: Age (being older) had an indirect negative association with people's perceived threat to the countryside through utilitarian attitude in Switzerland, Sweden and Spain.<br>*Gender*: Being a woman had a positive association with people's perceived threat to the countryside through utilitarian attitude in the Czech Republic, Spain and Switzerland. | |

## 4. Discussion and policy implications

This research aimed to identify and understand people's conservationist and preservationist environmental attitudes, how these relate to perceived threats to the countryside, and how these factors vary demographically and between countries. Given previous research, it was predicted that pro-environmental attitudes would be associated with greater perceived threat to the countryside, that this association would vary by age, gender, education level, between urban and rural dwellers, and with the frequency with which people visited the countryside.

### 4.1 Environmental attitudes

Preservationist attitudes were more prevalent than utilitarian attitudes in all five countries. Although the proportion of preservationist and utilitarian attitudes differed between countries, effect sizes were small. Consistent with previous research conducted in Portugal [32] and Ireland [74], preservationist and utilitarian attitudes were negatively correlated in four out of the five countries included in the research. Swiss citizens differed in this respect and appeared to have a more balanced distribution of the two attitudinal factors, with some believing that nature and natural species can be well protected whilst at the same time being used for human purposes. This is in keeping with existing research indicating the importance of societal demand to ecosystem management in Switzerland [75] and that the Alpine region, where Switzerland is located, is considered to have the best conservation status of all habitats in Europe [57]. This contrasts with the other countries, where preservationist and utilitarian attitudes appeared to be mutually exclusive. The observation that preservationist attitudes were lowest, and utilitarian attitudes highest, in Switzerland, implies a need for specific policies and communications to encourage people in Switzerland to shift towards more pro-environmental attitudes, potentially focused on the multi-functional nature of landscapes and the services which they deliver, accounting for the regulation and maintenance ES.

As expected, given previous research results [27, 28], environmental attitudes varied demographically within and between countries. Differences in utilitarian attitudes between demographic groups were unexpected. Whereas utilitarian attitudes to the environment were more common among men in the UK, utilitarian attitudes were more prevalent among women in the Czech Republic, Spain and Switzerland. Existing research in contrast, has been consistent in reporting that women tend to be more concerned about the environment [76, 77] and that pro-environmental, preservationist attitudes were more common among women [78]. Further research is required to understand these gender differences and how they relate to how the environment is used in these countries. Meanwhile, our findings imply a need for interventions to raise environmental awareness among men in the UK and among women in Czech Republic, Spain and Switzerland. Communication with people who hold utilitarian attitudes could focus upon reducing the psychological distance between the environment and individuals, or by considering recreational and provisioning functions of the countryside in relation to the activities in which they may engage [79]. That preservationist attitudes were more evident among men in Spain may be a consequence of greater cultural connectivity with the landscape (e.g. in the Dehesa), but requires further investigation.

As predicted [32, 33], preservationist attitudes were more common among older people in the UK, Sweden and Spain. Utilitarian attitudes were most prevalent among younger people in four countries (Spain, Sweden, Switzerland, UK). Young people therefore may be more likely than older people to view the countryside in recreational terms [27] and communication to encourage pro-environmental attitudes and engagement with land management initiatives should reflect this. This also underlines the need for tourism to consider sustainability in land use [15, 80]. Targeting communication to encourage preservationist attitudes among younger

people in these countries, for example, *via* social media [37], educational programmes in schools and colleges, and through embedding more frequent contact with nature, could increase pro-environmentalism [81], assist in promoting sustainable land management and render pro-environmental policies more effective [82].

Consistent with existing evidence for more positive environmental attitudes among those of higher socio-economic status [31], preservationist attitudes were more frequently observed among more educated respondents, but only in the UK. Respondents in the different countries also varied in the distribution of utilitarian attitudes with education level. As expected, given evidence for more positive environmental attitudes among those of higher socio-economic status (SES) [31], utilitarian attitudes were associated with having a lower education level in Spain. Contrary to our hypotheses, however, utilitarian attitudes were associated with a higher education level in the UK and Switzerland. This could reflect the large differences between countries in how people use the countryside and the degree to which people reported using the countryside for recreation and leisure (Czech Republic 49%; Spain 74%; Sweden 58%; Switzerland 63%; UK 79%). Interventions to raise awareness of environmental preservation therefore should target those who spent less time in education in the UK.

Preservationist attitudes were associated with more frequent visits to the countryside in Switzerland and the UK, suggesting that land management strategies to encourage people to spend time in the countryside could result in more pro-environmental attitudes. Such initiatives would be provided that sustainability measures were embedded within activities offered, the facilities provided, and that the environment and associated biodiversity were protected [15]. Such an approach has already been adopted by some agri-environment schemes such as the Environmental Land Management Scheme (ELMS) in the UK [83]. Utilitarian attitudes, however, were unrelated to the frequency with which people visited the countryside.

Neither preservationist nor utilitarian attitudes differed between rural and urban dwellers in any of the five countries. This was unexpected given previous evidence for more pro-environmental attitudes among rural dwellers [84] and evidence regarding polarised views on the prioritisation of resources in the countryside between rural and urban dwellers [28]. This may reflect the present study's use of validated measures on large representative samples rendering the findings reliable at scale.

## 4.2 Perceived threat to the countryside

Overall, the highest rated perceived threat was future socio-economic uncertainty owing to a lack of younger farmers and farm successional issues. Farmers are more likely to be older in European countries [85] and this can become more problematic as farmers age, given that farm succession plans are linked to generational renewal as well as cultural, economic and policy drivers of decision-making [86]. This implies that citizens in all countries included in the study would be more likely to support policies aimed at encouraging young people to work in the agricultural sector. Similarly, threats posed by agricultural activities linked to environmental degradation were rated highly by citizens within all countries, suggesting that policies aimed at reducing the environmental impacts of farming (e.g. through introduction of subsidies of agro-ecological practices [87] might be evaluated positively.

As expected, given previous cross-national research [36], there were between country differences in perceived threat to the countryside. Perceived threats to the countryside were rated as greatest in the UK. Nine out of the fifteen threats, including all those related to environmental degradation, were rated higher in the UK than in any of the other countries. Of the five biogeographical regions included, the Atlantic region, where the UK is located, has the worst conservation status of habitats [57]. This poor status might have led to stronger public

concerns about environmental issues. Levels of perceived environmental threat associated with food production have also been found to be higher in the UK compared to China [46]. This could also be attributed to land-use, regulatory and political factors specific to the region, for example, BREXIT [88] and attitudes towards related governance and national policies. Spain scored highest on four perceived threats related to improper land use and socio-economic uncertainty. This reflects the outcomes of a recent consultation with stakeholders on countryside management which highlighted extensive perceived policy weaknesses in Spain [88]. Threat perceptions could also have been associated with socio-economic issues specific to the Spanish region such as those related to coastal development and over-tourism [89].

Perceived threats to the countryside that were associated with specific issues also varied across countries. For instance, the highest perceived threat was "bad behaviour of visitors" in the Czech Republic, Switzerland and the UK, and "lack of young farmers taking over farming" in Spain and Sweden. Concern for the sustainability of farming has also been indicated in a previous policy analysis [88] with stakeholders in Spain and Sweden arguing a need for policies to incentivise farming and to encourage more sustainable farming practices. We also found evidence to suggest that some issues associated with relatively lower perceived threat to the countryside could be context dependent. For example, the lowest perceived threat in the UK was "conversion of pasture or meadow to woodland" which could reflect efforts to promote tree planting and woodland generation as part of the UK government strategy to mitigate climate change and contribute to the delivery of net zero targets [90]. Although some environmental pressures are threatening the countryside and the ES to which it delivers, these may be currently perceived by citizens to be of relatively low threat. Land abandonment, for example, is an increasing problem in some regions in Spain, but continues to be perceived as low threat by citizens [91]. The *a-priori* qualitative research conducted as part of the SUPER-*G* project in these same countries, also indicated that the public held contrasting cross-national views on land use [28].

Although there were no demographic differences in perceived threat to the countryside in Switzerland and the UK, differences were observed between demographic groups in the other countries. Greater perceived threat to the countryside was higher among men in Sweden, and increased with advancing age in Sweden and Spain, and with education level in Czech Republic. This finding contrasts with previous research indicating greater environmental awareness among younger people [34, 35] or no association with age [92]. Demographic variation in perceived threat could reflect the relative salience of preservation and land use challenges faced in different regions and national variation in regulatory frameworks associated with recreation and agriculture [8]. Demographic differences may also vary between and within countries which differ in terms of their geographies and agri-ecology.

People who perceive threats to the environment, for example, related to climate change, tend to be more concerned about environmental issues [48]. Perceptions of environmental threats are amenable to change over time [49] and threat appeal communication can bring about greater concern for the environment [50, 51]. Understanding citizens' perceptions of threats to the countryside can help to prioritise and inform the content of social innovations relevant to tackling environmental sustainability challenges and which may garner greater citizen support [93]. Integration of social and ecological targets in policies and interventions may more effectively address sustainability targets [94], increase the societal "visibility" of environmental threats and hence increase awareness e.g. *via* an "availability" heuristic [95]. Threats that are perceived to be of greater magnitude [47] or to have closer geographical proximity to an individual [96] may be more likely to induce pro-environmental behaviours. Thus, efficacy of pro-environmental communication through, for example, social media or as part of (school, workplace, and public) educational programmes and interventions, can be facilitated by

inclusion of information about the magnitude and proximity of a perceived threat to the countryside.

## 4.3 Preservationist and utilitarian attitudes and perceived threat to the countryside

As predicted, given evidence that pro-environmental attitudes are associated with the perceived costs of environmental degradation [26] and environmental threat associated with production practices [46], having higher preservationist attitudes was associated with greater perceived threat to the countryside in all five countries. If preservationist attitudes are amenable to intervention through information provision [97], e.g. threat-appeal communication [37, 50], it may be possible to increase perceptions of threat and hence support of pro environmental behaviours [26], as well as the effectiveness of land management policies targeting multifunctional land use. Further, that preservationist attitudes mediated the relationship between greater perceived threat to the countryside and being older in the UK, Sweden and Spain, implies that older people may be most responsive to communication on environmental threat. At the same time there is a need for policies and intervention in these countries to raise awareness of environmental threats among younger people. Utilitarian attitudes were also associated with greater perceived threat to the countryside in four countries (Czech Republic, Spain, Sweden and Switzerland). It is possible that people with utilitarian attitudes to the environment, through increased use of the countryside and engaging in cultural ES, become more aware of threats to the countryside [74, 82].

Contrary to what might be assumed to be a psychological "conflict" between land use and conservation [13, 14], preservationist and utilitarian attitudes may not always be mutually exclusive, as in the case of Swiss citizens in this research. There may also be differences in the association between utilitarian and preservationist attitudes and specific types of perceived threat to the countryside. People with utilitarian attitudes may be more concerned than those with preservationist attitudes about threats that directly affect them as individuals [98], such as those negatively affecting the provisioning of food, water and raw materials (i.e. provisioning ES), or limiting opportunities for recreation and tourism (i.e. cultural ES). Accordingly, perceived threats which compromise regulatory and maintenance ES (e.g. water flow regulation, carbon storage, crop pollination, habitat provision), may be viewed as less of a threat [99]. Addressing potential disparities between individuals with stronger utilitarian or preservationist attitudinal tendencies may be important for future environmental risk management and effective communication strategies to promote pro-environmental behaviour. Utilitarian-focused interventions, for example, could promote land sharing as a means through which land multifunctionality can occur, such as by providing facilities for visitors that do not interfere with agricultural use. Preservationist-focused initiatives could emphasise the role of regulating and maintenance ES within landscapes subjected to land-use changes in working towards net-zero, for example through demonstration sites open to the public.

## 4.4 Strengths and limitations

Citizen Science, the involvement of the general public in environmental sustainability, is considered crucial to the achievement of SDGs [7]. Although the cross-sectional correlational survey design limits the degree to which we can infer causation from the results, the findings appear consistent with current knowledge regarding the specification of attitudinal and perceived environmental threat measures and potential links between them. Although fitting data does not prove causal assumptions, it renders them tentatively more plausible, particularly if later replicated [100]. Another limitation is that data were self-reported and as such subject to inaccuracy inherent in recall where behaviours were recollected.

Further, perceived threats to the countryside were assumed to be represented as a single factor when there was some evidence of differences between regions in the degree to which specific threats were related to environmental attitudes. For example, respondents with high scores on utilization in the UK rated potential bad behaviours of visitors as being less threatening. Respondents in the Czech Republic, in comparison, rated damage to land by dense livestock stocking highly, while those in Spain rated the lack of young farmers taking over farming as a greater threat. These differing perspectives between countries are consistent with protection motivation theory [42, 43], which holds that perceived threats are context dependent. According to protection motivation theory, both the extent to which an issue is perceived to be threatening together with perceived effectiveness of mitigating behaviours, is related to greater adoption of such behaviours [20, 101]. Provision of information has been shown to elevate perceived threat and coping efficacy in bringing about pro-environmental behaviour [44, 50, 51]. These differing views on perceived threats between countries may need to be considered when communicating with citizens about environmental issues and in encouraging actions to mitigate and manage threats. The research design we adopted did not link consideration of perceived threat with perception of mitigation strategies, which may represent a useful topic for future research.

The survey design was robust in terms of sample size and because respondents were quota sampled to be representative on gender, age, education and 50% rural/urban dwelling in all countries and in terms of the measures employed. The EAI has been previously validated in Europe [32, 54, 59, 102] and the 24-item version has been applied here with good reliability to assess environmental attitudes employing the 'conventional' two-factor (utilitarian/preservationist) structure [103, 104]. Despite proven reliability and validity [54], it is possible that the 24-item EAI by virtue of having fewer items is less comprehensive than the 36-item version [32]. The perceived threat to the countryside outcome measure, although unvalidated, also showed good reliability and was strengthened by having items derived from *a priori* qualitative research.

## 5. Conclusion

This study investigated the relationship between environmental attitudes and perceived threat. The analyses identified cross-national and demographic differences in preservationist and utilitarian attitudes and how they relate to perceived threats to the countryside. Strategies to encourage pro-environmental behaviour are most effective where individual and government regulation go hand-in-hand [105]. The link between preservationist attitudes and perceived threat in all countries is encouraging, assuming that pro-environmental attitudes can be enhanced through a combination of regulatory policy and communication strategies that raise awareness of threats to the countryside [97] and encourage collective agency to behave in ways that mitigate risk associated with specific threats. Given that preservationist and utilitarian attitudes were both associated with higher perceived threat to the countryside among those recruited in the Czech Republic, Sweden, Switzerland, and Spain, this may reflect a growing awareness of environmental threats by citizens within these countries, and also implies that similar policies along with risk management and communication strategies may be effective in addressing the UN SDGs in these regions. Demographic differences in environmental attitude were observed between countries, which was to be expected given they were selected to differ on biogeographical factors that could impact upon land use. To promote greater preservationist environmental attitudes, separate and demographically targeted communication strategies will be required for different countries. These data imply it may be possible to promote preservationist attitudes (and potentially preservationist behaviour) through communication

management strategies that raise awareness of environmental threats associated with SDG 2 zero hunger and sustainable agriculture, SDG 12 sustainable production and consumption, and SDG 15 protection and restoration of ecosystems and biodiversity.

## Supporting information

**S1 File. Cross-national item-specific comparisons.**
(DOCX)

**S1 Table. Results of mediating effects.**
(DOCX)

## Author Contributions

**Conceptualization:** Shan Jin, Lynn J. Frewer.

**Data curation:** Sophie Tindale, Victoria Vicario-Modroño, Simona Miškolci, Mercy Ojo, Pedro Sánchez-Zamora, Rosa Gallardo-Cobos, Paul Newell-Price, Martijn Sonnovelt, Erik Hunter.

**Formal analysis:** Brendan P. Bunting.

**Funding acquisition:** Lynn J. Frewer.

**Investigation:** Sophie Tindale, Lynn J. Frewer.

**Methodology:** Barbara J. Stewart-Knox, Brendan P. Bunting, Shan Jin, Lynn J. Frewer.

**Project administration:** Shan Jin.

**Supervision:** Shan Jin, Lynn J. Frewer.

**Writing – original draft:** Barbara J. Stewart-Knox.

**Writing – review & editing:** Brendan P. Bunting, Shan Jin, Victoria Vicario-Modroño, Simona Miškolci, Pedro Sánchez-Zamora, Rosa Gallardo-Cobos, Lynn J. Frewer.

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
