## [Decision Letter · Decision Letter 0]

2 Jul 2024

PONE-D-23-43077Citizen attitudes towards the environment and association with perceived threats to the countryside: evidence from countries in five European biogeographic zonesPLOS ONE

Dear Dr. Jin,

Thank you for submitting your manuscript to PLOS ONE. After careful consideration, we feel that it has merit but does not fully meet PLOS ONE’s publication criteria as it currently stands. Therefore, we invite you to submit a revised version of the manuscript that addresses the points raised during the review process.

We look forward to receiving your revised manuscript.

Kind regards,

Shenghua Xie

Academic Editor

PLOS ONE

Journal Requirements:

"The SUPER-G project (Grant Agreement No.: 774124) has received funding from the European Union Horizon 2020 Research and Innovation Programme. The views and opinions expressed in this paper do not represent the official position of the European Commission and is entirely the responsibility of the authors."

Reviewers' comments:

Reviewer's Responses to Questions

**Comments to the Author**

1. Is the manuscript technically sound, and do the data support the conclusions?

Reviewer #1: Yes

Reviewer #2: Yes

2. Has the statistical analysis been performed appropriately and rigorously? 

Reviewer #1: Yes

Reviewer #2: N/A

3. Have the authors made all data underlying the findings in their manuscript fully available?

Reviewer #1: Yes

Reviewer #2: Yes

4. Is the manuscript presented in an intelligible fashion and written in standard English?

Reviewer #1: Yes

Reviewer #2: Yes

5. Review Comments to the Author

Reviewer #1: Based on the sample of five European countries in different geographical environments, this paper uses the structural model to survey the citizen attitudes towards the environment and association with perceived threats to the countryside, which is significant for promoting public support for environmental protection policies. However, reading this manuscript, some apparent problems exist and need further improvement. The comments are as follows:

1. The introduction needs to summarize and refine the innovation of your article better. Compared with existing research, how innovative is the article? This description needs to be stronger in the introduction.

2. It is better to have a separate research area section to support the explanation and show the research area map with latitude and longitude information.

3. Figure 1 in the manuscript can be optimized further to show more research content and improve aesthetics.

4. Why did the author(s) not build the regression model equation and explain it?

5. The analysis of the discussion section is not deep enough, so it is suggested that the author(s) further enrich this section to highlight research contributions. At the same time, it should highlight the differences between previous studies.

6. The author(s) should improve the timeliness of references to the manuscript and cite more high-quality literature in recent years.

7. It is suggested that the author(s) revise the manuscript to improve the quality of the languages and readability with the help of a native English speaker.

Reviewer #2: The study discusses the environmental attitudes of citizens in five countries (Czechia, Spain, Sweden, Switzerland and the United Kingdom) in relation to perceived threats and used the Environmental Attitude Inventory (EAI-24) method to measure citizens' environmental attitudes. The selection of topics makes sense and falls within the scope of the journal, but the following suggestions exist and require further reflection:

1. Data

The data combines questionnaires from five countries, and it is good that gender, place of residence, education level, etc., are combined in the demographic data. However, whether the proportion of women living in urban or rural areas is also 50 percent (as it is for men) requires further explanation.

2. Framework

I see that the article puts a very simple framework at the end, and would suggest that this framework be added to the second part, and that a paragraph be added to describe the research design, methodology, and structure of the article.

3. Analysis and Discussion

The article starts with citizens with different environmental attitudes and compares the different outcomes of the five countries, but the analysis of the driving mechanism behind that outcome is insufficient, and further thought could be given to this aspect, and the discussion section needs to be further elevated to the height of the article. The advantages of having data across five countries are not fully realized, e.g., what are the influencing factors behind a given data result? Is the phenomenon specific to a particular country and why? How do different national circumstances (stage of development, economic structure, national spirit, policy trends, etc.) in different countries manifest themselves in the data results? The paper mentions the terrain represented by the different countries, Continental, Mediterranean, Boreal, Alpine, Atlantic, but there is no discussion of these factors in the analysis, and whether the citizens of the different countries have their environmental attitudes connected to the place.

4. Significance

The significance of the study is somewhat general and requires further thinking. The factors that contribute to the different situations in different countries are organic and integrated, and conclusions simply attributed to media campaigns and policies are insufficient and not specific enough. What efforts have countries with higher citizen environmental attitudes made? What lessons can be learned and improved?

6. PLOS authors have the option to publish the peer review history of their article (what does this mean?). If published, this will include your full peer review and any attached files.

Reviewer #1: No

Reviewer #2: No

---

## [Author Response · Author response to Decision Letter 0]

1 Aug 2024

We have uploaded the "Response to reviewers' comments" document. Thanks.

---

## [Decision Letter · Decision Letter 1]

3 Sep 2024

PONE-D-23-43077R1Citizen attitudes towards the environment and association with perceived threats to the countryside: evidence from countries in five European biogeographic zonesPLOS ONE

Dear Dr.  Jin,

Thank you for submitting your manuscript to PLOS ONE. After careful consideration, we feel that it has merit but does not fully meet PLOS ONE’s publication criteria as it currently stands. Therefore, we invite you to submit a revised version of the manuscript that addresses the points raised during the review process.

We look forward to receiving your revised manuscript.

Kind regards,

Shenghua Xie

Academic Editor

PLOS ONE

Journal Requirements:

Reviewers' comments:

Reviewer's Responses to Questions

**Comments to the Author**

1. If the authors have adequately addressed your comments raised in a previous round of review and you feel that this manuscript is now acceptable for publication, you may indicate that here to bypass the “Comments to the Author” section, enter your conflict of interest statement in the “Confidential to Editor” section, and submit your "Accept" recommendation.

Reviewer #1: All comments have been addressed

Reviewer #2: All comments have been addressed

2. Is the manuscript technically sound, and do the data support the conclusions?

Reviewer #1: Yes

Reviewer #2: Yes

3. Has the statistical analysis been performed appropriately and rigorously? 

Reviewer #1: Yes

Reviewer #2: Yes

4. Have the authors made all data underlying the findings in their manuscript fully available?

Reviewer #1: Yes

Reviewer #2: Yes

5. Is the manuscript presented in an intelligible fashion and written in standard English?

Reviewer #1: Yes

Reviewer #2: Yes

6. Review Comments to the Author

Reviewer #1: The author’s careful revision of the reviewers’ comments has brought the article to a standard of academic quality that meets the publication criteria of this journal. It is recommended that the article be accepted for publication.

Reviewer #2: The revisions to the article have improved considerably this time and basically meet the requirements for publication, with the following additional comments provided for reference

1. Please revise Figure 2 to increase the readability of the image.

2. check the language of the article

3. why don't I see the author's responses to the reviewers?

7. PLOS authors have the option to publish the peer review history of their article (what does this mean?). If published, this will include your full peer review and any attached files.

Reviewer #1: No

Reviewer #2: No

---

## [Author Response · Author response to Decision Letter 1]

5 Sep 2024

We have uploaded the "response to reviewers" document which has outlined how we addressed the comments from Reviewer 2.

---

## [Editor Report · Decision Letter 2]

12 Sep 2024

Citizen attitudes towards the environment and association with perceived threats to the countryside: evidence from countries in five European biogeographic zones

PONE-D-23-43077R2

Dear Dr. Jin,

We're pleased to inform you that your manuscript has been judged scientifically suitable for publication and will be formally accepted for publication once it meets all outstanding technical requirements.

Kind regards,

Shenghua Xie

Academic Editor

PLOS ONE
---

## [Editor Report · Acceptance letter]

30 Sep 2024

PONE-D-23-43077R2 

PLOS ONE

Dear Dr. Jin, 

I'm pleased to inform you that your manuscript has been deemed suitable for publication in PLOS ONE. Congratulations! Your manuscript is now being handed over to our production team.

Kind regards, 

on behalf of

Dr. Shenghua Xie 

Academic Editor

PLOS ONE